# Thyroid Hormone Signaling in Embryonic Stem Cells: Crosstalk with the Retinoic Acid Pathway

**DOI:** 10.3390/ijms21238945

**Published:** 2020-11-25

**Authors:** Mercedes Fernández, Micaela Pannella, Vito Antonio Baldassarro, Alessandra Flagelli, Giuseppe Alastra, Luciana Giardino, Laura Calzà

**Affiliations:** 1Department of Veterinary Medical Science, University of Bologna, Via Tolara di Sopra, 50, 40064 Ozzano Emilia, BO, Italy; mercedes.fernandez@unibo.it (M.F.); luciana.giardino@unibo.it (L.G.); 2Fondazione IRET, Via Tolara di Sopra, 41/E, 40064 Ozzano Emilia, BO, Italy; micaela.pannella@gmail.com; 3Interdepartmental Center for Industrial Research in Life Sciences and Technologies, University of Bologna, Via Tolara di Sopra, 41/E, 40064 Ozzano Emilia, BO, Italy; vito.baldassarro2@unibo.it (V.A.B.); alessandra.flagelli2@unibo.it (A.F.); giuseppe.alastra2@unibo.it (G.A.)

**Keywords:** embryonic stem cells, thyroid hormone, nuclear receptors, neuroectoderm, retinoic acid, nestin

## Abstract

While the role of thyroid hormones (THs) during fetal and postnatal life is well-established, their role at preimplantation and during blastocyst development remains unclear. In this study, we used an embryonic stem cell line isolated from rat (RESC) to study the effects of THs and retinoic acid (RA) on early embryonic development during the pre-implantation stage. The results showed that THs play an important role in the differentiation/maturation processes of cells obtained from embryoid bodies (EB), with thyroid hormone nuclear receptors (TR) (TRα and TRβ), metabolic enzymes (deiodinases 1, 2, 3) and membrane transporters (Monocarboxylate transporters -MCT- 8 and 10) being expressed throughout in vitro differentiation until the Embryoid body (EB) stage. Moreover, thyroid hormone receptor antagonist TR (1-850) impaired RA-induced neuroectodermal lineage specification. This effect was significantly higher when cells were treated with retinoic acid (RA) to induce neuroectodermal lineage, studied through the gene and protein expression of nestin, an undifferentiated progenitor marker from the neuroectoderm lineage, as established by nestin mRNA and protein regulation. These results demonstrate the contribution of the two nuclear receptors, TR and RA, to the process of neuroectoderm maturation of the in vitro model embryonic stem cells obtained from rat.

## 1. Introduction

Thyroid hormones (THs), both the active form triiodothyronine (T3) and the pro-hormone thyroxine (T4), play key roles in physical and mental development and maturation in all animal species, having multiple and different effects during embryonic, fetal and postnatal life. In humans, this role is well illustrated by the consequences of untreated clinical and subclinical hypothyroidism, which leads to an insufficient supply of maternal THs to the fetus [1]. When occurring in the early stages of pregnancy, maternal hypothyroidism has deleterious effects, in particular on central nervous system (CNS) development, resulting in intellectual disability [2], as dramatically illustrated by the clinical syndrome known as “cretinism” [3].

The cellular action of THs is exerted via genomic and non-genomic mechanisms [4,5]. Genomic effects are mediated by two classes of thyroid hormone nuclear receptors (NRs), TRα and TRβ, encoded by *THRA* and *THRB* genes [6]. The primary transcript for each gene can be alternatively spliced, generating four different thyroid hormone receptors (TRs) in humans, TRα1, TRα2, TRβ1 and TRβ2, belonging to the large family of ligand-dependent transcription factors. Each receptor contains three domains: the transactivation domain that interacts with other transcription factors to form complexes that repress or activate transcription; the DNA-binding domain that binds to sequences of promoter DNA known as hormone response elements, and a ligand-binding and dimerization domain at the carboxy-terminus. Following ligand binding, TRs heterodimerize mainly with retinoid X receptors (RXRs), a family of NRs consisting of three members, RXRα, RXRβ and RXRγ. The binding of the NR dimers to target genes results in either suppression or induction, depending on ligand presence and the recruited co-factors [7]. Moreover, a complex biochemical machinery consisting of TH membrane transporters (MCT8, MCT10), synthesis (D1 and D2) and degrading (D3) enzymes of the active form T3 are also present in almost all cells to guarantee an appropriate concentration of intracellular T3 [8]. Finally, extracellular TH receptors, namely integrin αvβ3, also mediate TH effects during development and adulthood [9].

Retinoic acid (RA) also plays key roles in early development. RA is a derivative of vitamin A, an essential nutrient which cannot be synthesized by the embryo and must be acquired from the maternal circulation through the placenta [10]. The cellular action of RA is also mediated by NRs belonging to the same superfamily as TRs. RA receptors (RARs), in fact, form heterodimers with RXRs and TRs, bind to RA-responsive elements (RAREs), and regulate expression of the target genes [11].

In humans, the onset of fetal thyroid function occurs at week 12 and is completed by 18 weeks of gestation, while vitamin A can not be synthesized by the embryo, which can instead produce the active form of RA. However, both THs and RA of maternal origin mediate very early developmental events. The maternal-fetal exchange of these factors begins though the transient structure yolk sac and the fetal exocoelomic cavity [12,13,14], after which the presence of different classes of TH membrane transporters in the placenta, such as organic anion transporting polypeptides, monocarboxylate transporters, and transthyretin for THs, together with the placental TH inactivating enzyme deiodinase 3 (D3), guarantee- the feto-maternal TH equilibrium [15]. The role of placental retinoic binding proteins (RBPs), transthyretin and other proteins (Strat6, CRBPI) as vitamin A trans-placental carriers is still debated [10].

Sporadic data indicates that pre-implantation embryos are also TH- and RA-sensitive. For example, THs are present in ovarian follicular fluid [16], TRs are expressed up to the blastocyst stage in the bovine preimplantation embryo [17], and the eight-cell morula and inner mass in human embryos express low levels of TRα and TRβ [18]. More generally, the in vitro exposure of bovine embryos to THs significantly increases blastocyst formation, hatching rates, increases the total cell count, and reduces apoptotic cells [19]. 

Our aim in this study was to explore the potential role of THs and RA in early pre-implant embryonic development, focusing on the possible genomic interaction between the TH and RA receptors. For this purpose, we used embryonic stem cells (ESCs) derived from the inner cell mass of rat blastocyte obtained 4.5 days *post coitum*. These cells, which are characterized for the ESC *bona fide* properties [20], can be used as a cell line, and cultured in different conditions as a monolayer or 3D structure (spheroids and embryonic bodies). 

## 2. Results

### 2.1. RESC Culture Conditions Determine Culture Type and Cell Properties

ESCs cultured in vitro mimic all the different and timely regulated cellular and molecular processes necessary for the success in blastocyte pre-implantation [19]. To reproduce a number of key processes of this stage, we cultured RESC in different conditions, medium compositions and culture surfaces, yielding RESC for growth in different cell aggregations, exhibiting different cellular and molecular properties. In particular, in the presence of LIF and bFGF, cells growing on ultra-low attachment (ULA) plates proliferated as clusters (CL) and progressively increased in size (Figure 1B). When the CL reached a diameter of about 300 µm (after 6–7 DIV approximately), they were mechanically split, after which there were two possible approaches: (i) SCML-new medium on standard surfaces treated for cell adhesion, promoting cell proliferation, where cultures exhibited a mixed population of floating clusters (mCL) and single cells (mSC) attached to the surface (Figure 1C); or (ii) SCM without LIF and with bFGF (-LIF/+bFGF) on ULA surfaces. Following this second approach, cells proliferated as large spheroidal cell aggregate structures known as embryonic bodies (EB; Figure 1D) [20,21]. EB induction took 7 days of culture and represented the first step toward three-layer differentiation. When single cells obtained from the EB dissociation were then cultured onto coated surfaces in SCM medium without mitogens, they proliferated and differentiated (Figure 1E). Culture medium contained 10% of serum in all aforementioned conditions (FBS-ES).

RESC properties in the different culture conditions were investigated in terms of pluripotency (*Oct4* gene expression and Oct4 protein), inner cell mass differentiation (*Afp* and *Stella* gene expression), and cytoskeleton and extracellular matrix (ECM) protein expression (F-actin and laminin). Although pluripotency marker *Oct4* was expressed in all culture conditions, the highest *Oct4* expression was observed in the clusters obtained from the mixed population (mCL) grown in SCML-new medium in the presence of LIF and bFGF (Figure 2A). *Stella* mRNA (Figure 2B) and *Afp* gene expression level strongly increased in EB (*Stella* five times; *Afp* around 120-fold CL) compared to all the other cell populations (Figure 2C). Interestingly, the expression of these three genes in clusters from the mixed population (mCL), which grew in floating conditions, was higher than the mSC, which grew attached to the plate surface.

We also investigated the expression of marker proteins in the two culture conditions, which provided the extreme results at mRNA level, i.e., pluripotency-supporting conditions (CL in SCML-new medium and ULA surfaces) and differentiation-promoting conditions (EB in SCM -LIF/+bFGF medium and ULA surfaces). To achieve this, CL and EB cryosections were prepared and processed for ICC as described. Cells displaying *Oct4* immunostaining were observed in CL and EB (Figure 2D,G). The cytoskeleton F-actin (Figure 2F,I) and base membrane laminin (Figure 2E,H) proteins were both expressed in CL and EB.

### 2.2. Nuclear Receptors and co-Regulator Expression in the Different Culture Conditions

We then analyzed the gene expression of the entire set of NRs (47 in rat) and co-regulators in the four culture conditions reflecting the differentiation steps using PCR array technology, (see Appendix A for the full list of genes and fold regulations). This profile created a clustergram consisting of a cluster containing three groups (mCL, CL and mSC) with mCL and CL in the same sub-cluster, and the EB cells in a separate cluster (Figure 3A).

Of the total of 84 genes of interest, 22 genes were upregulated (>2 fold increase) in EB compared to CL (*Brd8*, *Esr1*, *Esr2*, *Esrrb*, *Gper1*, *Hnf4a*, *Med12*, *Med24*, *Ncoa2*, *Ncor1*, *Nr0b2*, *Ne1d1*, *Nr1h4*, *Nr1l3*, *Nr2f1*, *Nr2f2*, *Nr5a1*, *Ppara*, *Pparg*, *Rarb*, *Rbpjl*, *Rora*), and one was downregulated in both mCL and mSC compared to CL (*Nr1i1*). Three genes were downregulated in mSC (*Rxrg*, *Thrb* and *Vdr*) compared to CL, and *Vdr* was also downregulated in mCL. However, all of them increased expression in EB (Figure 3B).

Analyzing the entire list of the differentially expressed genes by pathway enrichment (GeneCodis 4.0), 21 of the 25 inputs were recognized by KEGG pathway analysis, showing TH signaling to be the most represented pathway (Figure 3C, Appendix A).

### 2.3. TH Signaling in RESC in the Different Culture Conditions

To analyze the expression of the regulators and mediators of TH signaling more in-depth, we then studied the expression level of the TH-converting enzymes (deiodinases D1 and D2, converting inactive T4 to the active T3; deiodinase D3, the inactivating enzyme converting T4 to rT3 and T3 to T2 [4]); TH transporters (MCT8, MCT10), and TH nuclear receptors (TRα1, TRα2, TRβ).

Overall, RESC expressed TH signaling genes and proteins (Figure 4). More specifically, D1 and D2 showed higher mRNA levels in the clusters of the mCL mixed population compared to CL, whereas they were downregulated in the mSC mixed population of single cells (Figure 4A,B), while EB express higher D2 levels compared to CL. Interestingly, the D3 mRNA enzyme was significantly upregulated in both mCL and mSC populations (Figure 4C).

The TH transporters MCT8 and MTC10 were also expressed in RESC; EB in particular showed the higher gene expression of both transporters (Figure 4D–H). The expression of both transporters was also significantly lower in mSC compared to CL and mCL. TRα1 was equally expressed in all RESC growing conditions (Figure 4E), whereas TRα2 and TRβ mRNA levels were higher in EB (Figure 4F,G). D3, TRα1 and TRβ immunostaining of CL and EB cryosections is shown in Figure 4 panels I–N. As the figures show, the D3 protein was present in the cell cytoplasm (Figure 4I,L) whereas TRα1 and TRβ were also distributed throughout the nucleus (Figure 4J,K,M,N).

### 2.4. TH Signaling in Germ Layer Lineage: Focus on Neuroectodermal Differentiation

To explore the potential role of TH signaling in neuroectodermal induction, the EBs were split, and the resulting single cells cultured onto gelatin coated coverslips in SCM medium without mitogens and in the presence of 1 μM RA. The neuroectoderm lineage of RESC was initially assessed using the neuroectodermal marker nestin, and representative images of the cell cultures at 5 and 10 DIV are shown in Figure 5A–D. At both analyzed time points, the number of nestin-positive cells was significantly higher in RA-exposed compared to RA-unexposed cells (Figure 5E,F) (5 DIV: 32.89 ± 2.75 vs. 50.83 ± 2.66; 10 DIV: 32.26 ± 5.97 vs. 88.85 ± 3.1). In addition, the number of nestin-positive cells under RA-exposure at 10 DIV was approximately 1.7 times that at 5 DIV. RA-treated nestin-positive cells showed a more differentiated morphology than spontaneously differentiated cells at both 5 and 10 DIV, as indicated by the shape of the cells and by the numerous elongations extending from the cell bodies.

To explore the possible role of TH signaling in RA-induced neuroectoderm lineage, we used the pharmacological tool TR antagonist 1-850, a selective and high-affinity TR antagonist which blocks T3-mediated interaction of TRα and TRβ with nuclear receptor coactivator. This molecule prevents TR-mediated genetic effects, without affecting retinoic acid receptor α (RARα) activity [22]. RESC were exposed to TR antagonist 1-850 in different culture conditions as illustrated in Figure 6A: (a) treatment lasted throughout all differentiation stages, from CL to differentiated single cells; (b) cells were treated only when cells grew as CL; (c) cells were treated during the EB induction period only; (d) cells were treated during the differentiation phase; and (e) cells were not treated at all, thus representing the experimental control group.

We counted the nestin-positive cells after 10 DIV of differentiation, and results are shown in Figure 6B–D. We first confirmed that RA is a key signaling molecule to induce neuroectodermal differentiation (Figure 6B), in fact less than 20% of cells expressed nestin when RESC were differentiated without RA, compared to 40% in the presence of RA. TR antagonist 1-850 decreased the percentage of nestin-positive cells in culture conditions (c) and (d), corresponding to TR antagonist 1-850 exposure during EB induction or during differentiation as single cell (Figure 6D), in a similar % effect. As a result, no differentiated single cells survived in condition a), in which the treatment lasted throughout all culture phases.

To confirm this result, and to explore the gene expression level, we treated single cells derived from EB with 1 μM RA, 10 μM TR antagonist 1-850, or both for 7 days, performing treatments every other day. As shown in Figure 6E, RA induced an increase in the expression of nestin mRNA of about 3-fold, whereas the TR antagonist 1-850 completely abolished the RA effect.

In order to investigate if TR antagonist 1-850 treatment may have an effect in the expression of TR-regulated genes, we studied the expression of Krüppel-like factor 9 (Klf9) mRNA, which is a TR-regulated gene [23,24]. We observed a significant increase of klf9 mRNA in the RA treated group of cells (Student’s *t* test, *p* = 0.0111), however no significant effect was observed in the group of cells treated with TR-antagonist 1-850 (Appendix A).

## 3. Discussion

During very early development of the embryo, before the onset of fetal thyroid gland function, THs of maternal origin reach the embryo through the feto-placental unit. However, emerging evidence indicates that THs may play an important role in very early embryonic development, for example forming part of the molecular machinery governing implantation from the maternal side. TRα1 and TRβ1 are expressed during the mid-luteal phase in glandular and luminal epithelium, increasing concomitantly until endometrial receptivity is established [25].

Very few studies have explored the potential role of TH signaling prior to implantation, i.e., at the morula-blastocyte stage. The little existing evidence suggesting that this developmental window may also be TH-sensitive include the expression of TRα1 and TRβ1 in the bovine cumulus cells, TRα expression in immature oocytes, and the improvement of bovine embryo development kinetics when oocytes are exposed to THs in vitro [26]. In bovine preimplantation embryos, TRs are expressed up to the blastocyst stage [17], as in human embryos, where the cells of the inner mass express low levels of TRα1 and TRβ1. THs stimulate mitochondria replication and energy production in human preimplantation embryos, by switching the glycolytic metabolic pathway to oxidative phosphorylation, and regulate the transcriptome [27].

The effect of THs on early development may be related to general cellular mechanisms (e.g., cell metabolism), but also to specific effects on cell differentiation and lineage. In this study, we explored this latter possibility, using in vitro systems based on ESCs. Since the 1980s, when in vitro methods for culturing ESCs first became available, great efforts have been dedicated to identifying the molecular mechanisms which contribute to the maintenance of their bona fide properties, self-renewal and pluripotential capabilities, including genetic and epigenetic factors and signaling pathways. In this study, we used a line of ESC obtained from rat (RESC), which had already been characterized for ESC bona fide properties [20]. The RESC populations CL, mCL and mSC were cultured in the presence of LIF, while EB were cultured in medium without LIF, which determines fate changes toward the three-layer lineage commitment. Notably, these culture conditions retained pluripotency marker expression, as indicated by *Oct4*, while *Stella* and *Afp*, both related to the three-layer lineage commitment [28,29], increased at the EB stage. These in vitro changes may resemble early in vivo embryo development, therefore this RESC model is a good candidate for the study of certain preimplantation embryo events.

We demonstrated that (i) in vitro RESC expressed nuclear receptor genes, in particular TRs and co-regulators; (ii) a robust switch toward increased expression of TH signaling molecules including synthesis and degrading enzymes and membrane transporters was observed at the EB stage; and (iii) THs played a key role in the neuroectodermal lineage induced by RA, since TR antagonists impaired RA-induced neuroectodermal differentiation.

Early embryos are highly sensitive to the environment under in vitro culture conditions, and the culture conditions can alter the gene expression pattern [30]. Both culture conditions we used to grow the RESC containing a very low T3 concentration (SCML-new about 300 pg/mL corresponding to 0.44nM; SCM about 170 pg/mL corresponding to 0.25 nM), values similar to the physiological T3 concentration in oviduct and endometrial tissue (0.67 nM) [19]. In fact, when T3 is used in vitro as a pharmacological tool, concentrations are usually 10 to 100 times higher [27,31]. Moreover, in studies demonstrating that TH exposure improves embryonic development by increasing the number and quality of formed embryos as well as the hatching rate, the culture media was supplemented with T3 at a concentration of nearly 74 nM [19]. The T3 concentration able to induce lineage effects (oligodendroglial lineage from neural stem cells), was 50 nM [32,33]. Overall, analysis of the expression of genes and proteins related to TH signaling showed that the EB stage was more responsive to TH internalization (higher TH transporter expression) and trigger of the genomic effect (higher TR expression). Deiodinase expression was also imbalanced in favor of higher synthesis of the active form T3 (higher expression of D1 and D2 and lower expression of D3). Finally, the RA- and TH-mediated pathways seemed to be more highly represented, therefore we can speculate that a default genetic program establishes embryonic cell sensibility to THs of maternal origin. This hypothesis is supported by the fact that T3 concentration in the culture media across the expression switch (mSC to EB) did not change significantly, being 0.44 or 0.25 nM.

We then investigated whether the TH-priming of RESC was also implicated in lineage specification, and not simply limited to general metabolic functions of these cells [27]. In fact, at blastocyte stage, which is mimicked by EB, gastrulation occurs, and the three-layered embryonic structure is established. We focused on neuroectodermal specification, which is driven by RA, as extensively described in vivo and in vitro [34,35,36] and as also confirmed for RESC used in this study by the nestin marker [37]. RA is one of the most important extrinsic morphogens, modulating ESC differentiation into various cell types in a time- and concentration-dependent manner [36,38,39]. Cross-talk between TH and RA signaling has been documented under physiological conditions. In particular, unliganded TRα has an inhibitory effect on the RA response in mouse embryonic stem cells, inhibiting RARβ expression and modulating RA-stimulated neural differentiation [40]. We first exposed RESC at all in vitro stages to both RA and the TR antagonist 1-850. The RA concentration we used for neuroectodermal induction (1 μM) was within the range of concentrations used in other studies performed in EB from mouse ESC (from 10^−8^ to 10^−6^ M) [34,36,39]. We demonstrated that as soon as EB takes place, e.g., the three-layer induction stage of increasing complexity [35,41] which corresponds to experimental conditions (c) and (d), RA cellular action requires TRs, suggesting that either RAR-TR nuclear receptor heterodimers or distinct, but simultaneously activated down-stream RAR and TR mechanisms are required for neuroectodermal induction. Future experiments will be performed to further investigate the role of RA and TH regulated pathways in this critical developmental stage.

## 4. Materials and Methods

### 4.1. Cell Culture

RESC were derived from the 4.5 days *post coitum* blastocyst as already described [20]. RESC-CL were grown without MEF in ULA (ultra-low attachment) plates in stem cell medium containing LIF supplemented with MEF conditioned medium (SCML-new) prepared as follows: Knockout DMEM (Dulbecco’s Modified Eagles’ Medium ES cells, Gibco, Waltham, MA, USA) supplemented with 50% MEF-CM (EmbryoMax^®^ MEF Conditioned Media, Merck-Millipore Burlington, MA, USA), 5% FBS-ES (Fetal Bovine Serum Embryonic Stem cells tested, Euroclone, Milan, Italy), 0.1 mM MEM/NEAA (Gibco), 0.1 μM β-mercaptoethanol, Nucleoside Mix (8 μg/mL adenosine, 7.3 μg/mL cytidine, 8.5 μg/mL guanosine, 2.4 μg/mL thymidine, 7.3 μg/mL uridine, all from Sigma-Aldrich, Burlington, MA, USA), 100 U/mL penicillin, 100 μg/mL streptomycin (Thermo Fisher Scientific, Waltham, MA, USA), 2 × 10^3^ U/mL LIF (leukemia inhibitor factor, Merck-Millipore, Burlington, MA, USA), 10 ng/mL bFGF (from Gibco, freshly added every 48 h). Cells were mechanically split when they reached a size around 300 μm. Clusters from passage 54 to 56 (P54–56) were used for EB induction. Briefly, clusters were split and the resulting cells were grown in ULA plates in SCM without LIF but in the presence of bFGF. SCM medium was composed of DMEM:F12 (Dulbecco’s Modified Eagle Medium: Nutrient Mixture F-12, Gibco), 10% FBS embryonic stem cell quality, 0.1 mM MEM/NEAA, 0.1 μM β-mercaptoethanol, Nucleoside Mix, 100 U/mL penicillin, 100 μg/mL streptomycin, 4 ng/mL bFGF (freshly added every 48 h).

For the differentiation studies, EB at 7 DIV were split and the resulting single cells were cultured in SCM medium without mitogens (no LIF, no bFGF) to allow them to differentiate. Cells were seeded at a density of 1−5 × 10^4^ cells/cm^2^ and treated for 5 to 10 days, changing the medium every 3–4 DIVs. For ICC studies, single cells were plated onto 0.1% gelatin coated coverslip, whereas for gene expression studies on single cells, no surface coating was performed.

### 4.2. Cell Treatments

The thyroid hormone receptor antagonist (1-850) (Cayman Chemical, Ann Arbor, MI, USA), 10 μM, was used to block T3 binding to both TRα and TRβ [22]. Retinoic acid (1 μM, all-*trans* RA, CAS N. 302-79-4, Sigma, Burlington, MA, USA) was used to induce neuroectoderm lineage differentiation. Both TR antagonist and RA were added to the culture medium every other day.

### 4.3. RT^2^ PCR Arrays

To perform the PCR arrays, total RNA isolation from the RESC populations CL, mCL, mSC and EB was performed using the RNeasy Micro kit (Qiagen, Milan, Italy) according to manufacturer’s instructions. Total RNA was quantified (Nanodrop 2000 spectrophotometer, Thermo Scientific, Waltham, MA, USA) and cDNA was produced using the RT^2^ First Strand Kit (Qiagen) according to manufacturer’s instructions. For each condition, a pool was created from independent experiments with a total of 1 µg of RNA per group: CL, mCL, mSC, and EB.

The PCR array for rat nuclear receptors and coregulators was used (PARN-056Z; Qiagen) in combination with the RT^2^ SYBR Green qPCR Mastermix (Qiagen), and the online software GeneGlobe (Qiagen) was used to analyze data and produce the graphical representations. GAPDH and β-actin were used as housekeeping genes, according to the software instructions. A ∆Cq of 37 was selected as cut-off for non-expressed genes.

All arrays passed the quality test with regard to reproducibility, positive control of PCR reaction, and genomic DNA contamination. Data was analyzed as ΔCq to build the clustergram and fold of regulation compared to a control group (CL). The differentially expressed genes (*n* = 25) were used as inputs for the pathway enrichment analysis using the GeneCodis software (v 4.0) and the KEGG algorithm.

### 4.4. RNA Isolation, Reverse Transcription and Semi-Quantitative Real-Time PCR

To perform the gene expression studies using classical qPCR, total RNA isolation was performed using the RNeasy Micro kit (Qiagen) and quantified as described above (Nanodrop 2000 spectrophotometer, Thermo Scientific). First-strand cDNA was prepared using the iScript™cDNA Synthesis Kit (BioRad, Hercules, CA, USA) according to manufacturer’s instructions. A no-reverse transcribed RNA sample (no RT control) was also prepared as control.

Semi-quantitative real-time PCR reactions were performed in a final volume of 20 μL consisting of 1× SYBR Green qPCR master mix (BioRad, Hercules, CA, USA) and 0.5 μM forward and reverse primers using the CFX96 real-time PCR system (BioRad). Control samples (no RT and no template control) were processed for each pair of primers used. Details of the primer sequences is included in Table 1. All primers used were designed using Primer Blast software (NCBI, Bethesda, MD, USA) and synthesized by IDT (Coralville, IA, USA). GAPDH was used as housekeeping gene.

PCR reactions were performed through the following steps and thermal profile: (1) denaturation step (98 °C, 3 min); (2) amplification (95 °C for 10 s and 60 °C for 60 s), 40 cycles; (3) melting curve of the amplified products (55 °C to 95 °C, ∆t = 0.5 ° C/s). The specificity of PCR reactions was evinced from the single peak obtained after performing the melting curve. Primer efficiency values for all primers were 95–102%, therefore the 2(−∆∆Ct) method was used to perform the analysis.

### 4.5. Immunocytochemistry

Immunocytochemistry (ICC) was used to study the protein expression of RESC in floating growing cells (CL and EB) and in single cells (SC) obtained from EB growing as monolayer on a coated surface. Detailed descriptions of the primary and secondary antibodies used, working dilutions and supply companies are given in Table 2.

*CL and EB cryosection ICC*: RESC growing in floating conditions (CL and EB) were processed for ICC following the procedure already described [42] with in-house modifications. In brief, cells were centrifuged (400× *g*, 5 min) at confluence and the supernatant was discarded. The cell pellet was incubated with 4% paraformaldehyde (PFA, in 0.1M Sørensen phosphate buffer, pH 7.4) for 30 min at RT, then washed with PBS for 5 min. Cells were then incubated with increasing % sucrose solutions (10-20-30) for 30 min each at RT. After removing as much of the sucrose solution as possible, cells were initially embedded with OCT (Optimal Cutting Temperature) compound, then dry ice frozen and stored at −80 °C until processing. 14 μm sections were prepared and mounted onto 200 mg/mL poly-l-lysine coated glass slides with a Cryostat (LEICA CM1950, Leica Biosystems, Milan, Italy). To perform the ICC, cryosections were air dried for 1 h at RT, then hydrated with PBS for 30 min at RT. After the blocking step (PBS/2% BSA/0.3% Triton-X 100), sections were incubated with primary antibodies at 4 °C overnight. After PBS washes (three times, 5 min each) and incubation with secondary antibodies for 30 min at RT, sections were incubated with the nuclear dye Hoechst 33258 (1 μg/mL in PBS/0.3% Triton-X 100) for 20 min at RT. Sections were finally washed with PBS (three times, 5 min each) and mounted in phenylenediamine solution (glycerol: PBS, 3:1, containing 0.1% 1,4-phenylenediamine, *w*/*v*).

*Single cell ICC:* cells were fixed with 4% PFA solution (30 min at RT) and washed three times with PBS for 5 min. After incubating with blocking solution (PBS/2% BSA/0.3% Triton-X 100), cells were incubated with primary antibodies at 4 °C overnight. After PBS washes (three times, 5 min each) and incubation with secondary antibodies for 30 min at RT, cells were incubated with Hoechst 33258 (1 μg/mL in PBS/0.3% Triton-X 100) for 20 min at RT. After washing, cells were mounted in phenylenediamine solution. Control experiments with the secondary antibodies alone were always carried out in parallel (results not shown). Staining was observed using an Eclipse E600 microscope (Nikon, Minato, Tokyo, Japan).

### 4.6. T3, T4 Quantification

T3 was quantified in the two culture media used, SCML-new for CL, mSC and mCL populations, and SCM for the EB and single cells obtained from them. The xMAP Luminex technology and the Rat Thyroid Magnetic Bead Panel—Endocrine Multiplex Assay kit (RTHYMAG-30K, Merck Millipore) were used, according to manufacturer’s instructions with appropriate in-house modifications. MAGPIX instrument and xPONENT 4.2 software were used to read the bead fluorescence and perform the analysis.

### 4.7. Statistical Analysis

Data is expressed as mean ± sem. Statistical analysis was performed with anova-1 way and tukey’s multiple comparison test to compare the different experimental resc groups. Student’s *t*-test was used to compare the two experimental groups. Significance was set at *p* < 0.05.

## 5. Conclusions

We demonstrated that embryonic stem cells derived from rat blastocytes progressively increased TH signaling genes in vitro depending on culture conditions, leading to “embryonic bodies” and in vitro 3D structures resembling blastocytes. Moreover, TH nuclear signaling is essential to allow RA-mediated neuroectodermal differentiation.

## Figures and Tables

**Figure 1 ijms-21-08945-f001:**
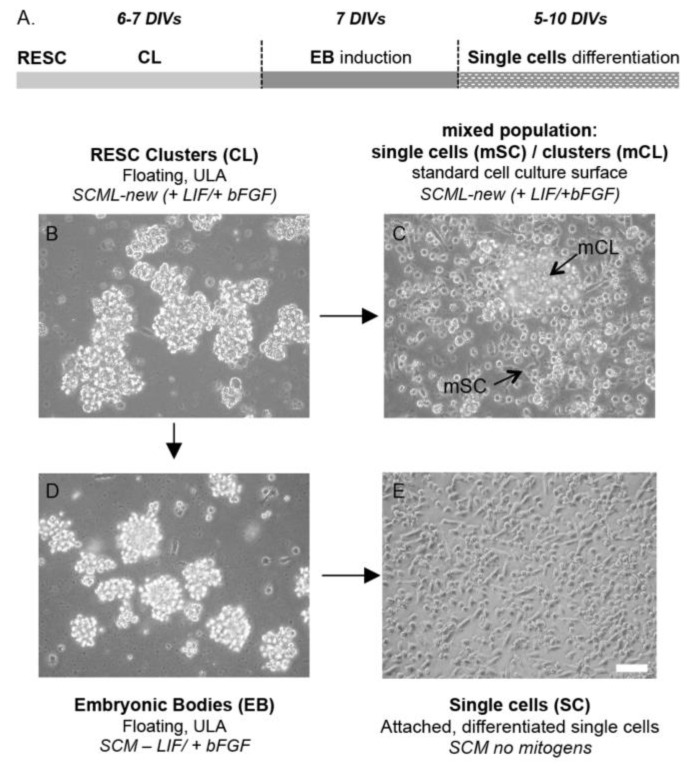
RESC populations and culture conditions. (**A**). Experimental and temporal approach to obtain the different RESC populations. (**B**–**D**). Representative images of cells cultured in different medium compositions and culture conditions are shown in the figure. When RESC grow in SCML-new medium (+LIF/+bFGF) on a ULA surface, cells proliferate as CL, increasing in size over the days in culture (**B**). Cells resulting from split CL can be either cultured in SCML-new medium (+LIF/+bFGF) on a standard culture surface (Nuclon Delta treated), giving a mixed population of floating clusters (mCL, black arrow in the picture) and attached single cells (mSC, black arrow in the picture) (**C**), or cultured in SCM medium -LIF/+bFGF on a ULA surface, thus inducing the formation of EB (**D**). Single cells obtained from the split of EB and cultured on a coated surface spontaneously differentiated (**E**). Scale bar 100 μm. Abbreviations: bFGF, basic fibroblast growth factor; EB, embryoid bodies; CL, clusters; LIF, leukemia inhibitory factor; mCL, mixed population clusters; mSC, mixed population single cells; SC, single cells; SCM, stem cell medium; SCML, SCM with LIF; ULA, ultra-low attachment.

**Figure 2 ijms-21-08945-f002:**
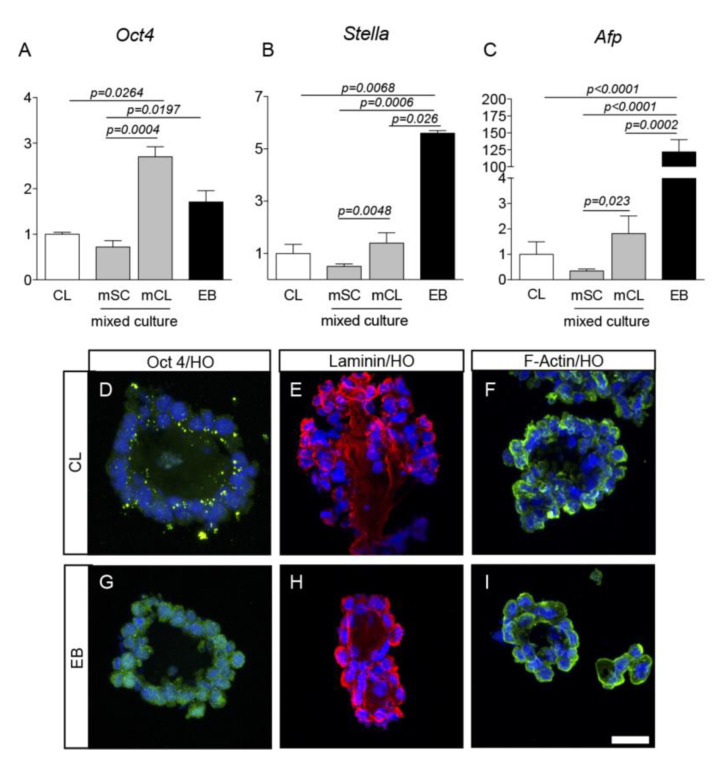
RESC pluripotency and differentiation capabilities. The expression of pluripotency and inner cell mass differentiation markers was studied in RESC: *Oct 4* mRNA (**A**) and protein (**D**,**G**); *Stella* mRNA (**B**); Alpha fetoprotein mRNA (**C**); Laminin (**E**,**H**) and F-actin **(F**,**I**) proteins. Gene expression was assessed by semiquantitative QPCR and the 2^(−∆∆Ct)^ method, using CL as reference group. The graphs show mean values ± SEM of *n* = 3 independent experiments. Samples were processed in duplicate. Statistics: One-way ANOVA and Tukey’s post test were performed. Significance was set at *p* < 0.05. Representative images of CL and EB cryosections stained with the specified antibodies and the nuclear colorant Hoechst 33258 are included in the figure. Scale bar 25 μm. Abbreviations: Afp, alpha fetoprotein; CL, clusters; EB, embryoid bodies; F-actin, filamentous actin; HO, Hoechst; Oct 4, octamer-binding transcription factor 4; mCL, mixed population clusters; mSC, mixed population single cells.

**Figure 3 ijms-21-08945-f003:**
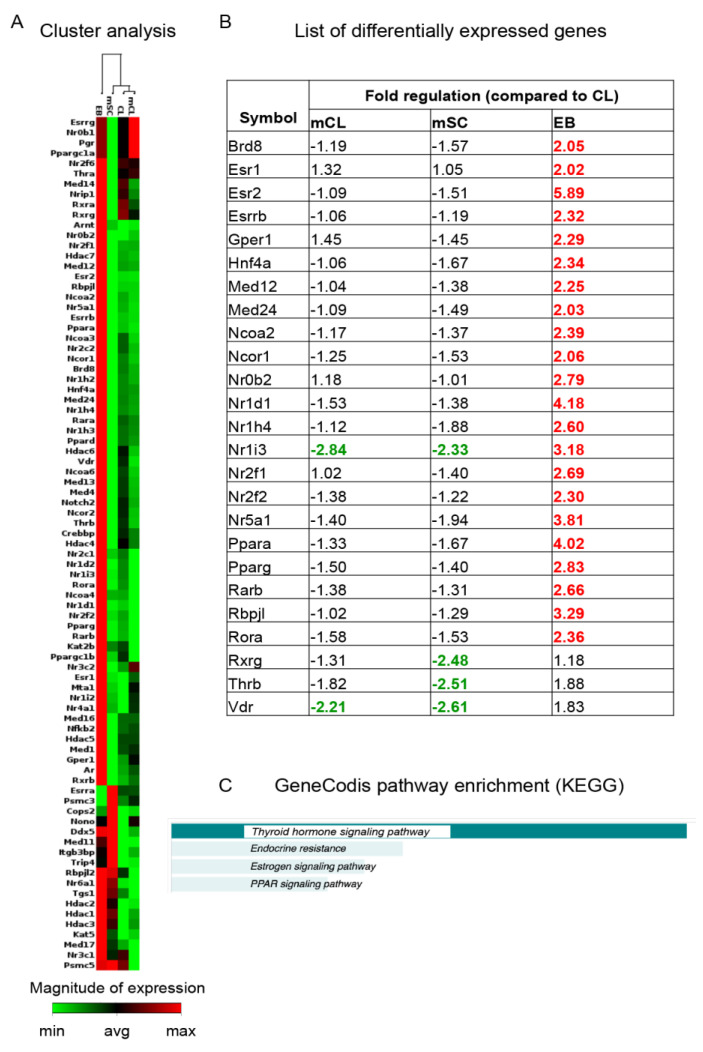
Nuclear receptor and co-regulator expression. The expression of all 47 NRs and 37 co-regulators were analyzed by RT2 PCR array technique and represented as a ΔCq heatmap for each gene in a clustergram showing the maximum (red) and the minimum (green) expression for each gene in the various groups (**A**). The fold of regulation value for each gene was calculated using CL as a control group, and genes showing more than 2-fold regulations are shown (**B**). Red color is for genes that are up-regulated whereas green color is for genes that are down-regulated. Bold numbers indicate fold of changes ≥ 2, both as positive and negative value. These genes were used as inputs for the pathway enrichment analysis using the GeneCodis software, and the results of the analysis is shown (**C**). Abbreviations: CL, clusters; Cq, threshold cycle; EB, embryoid bodies; mCL, mixed population clusters; mSC, mixed population single cells.

**Figure 4 ijms-21-08945-f004:**
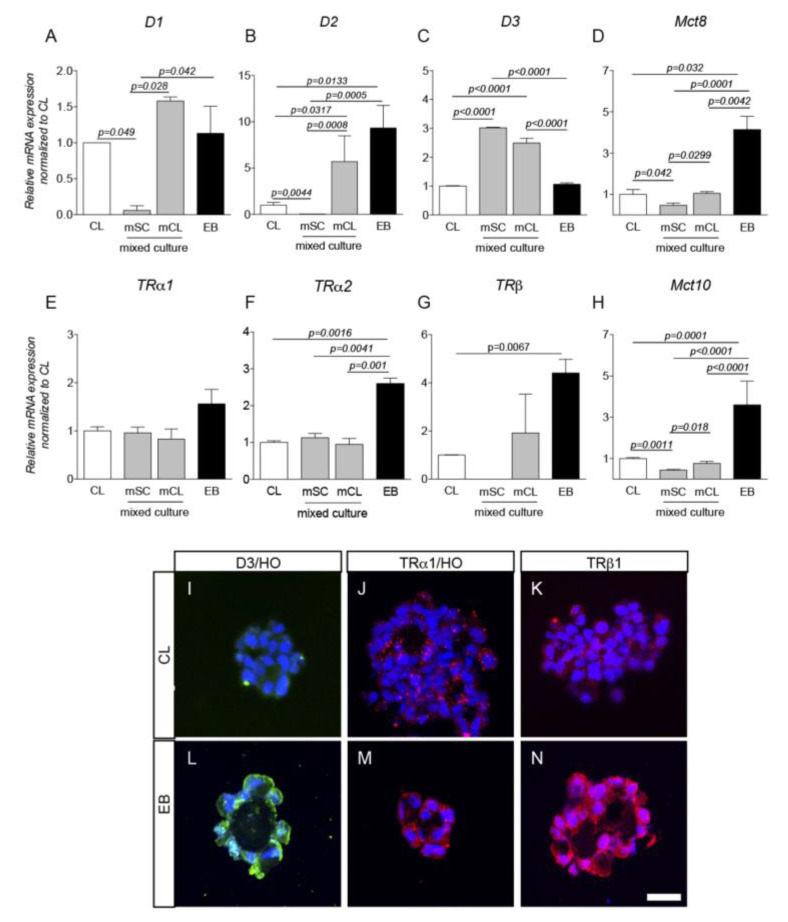
RESC and Thyroid Hormone. RESC gene expression of TH converting enzymes D1, D2, D3 (**A**–**C**), transporters MCT8, MCT10 (D,H) and receptors TRα1, TRα2 and TRβ (**E**–**G**) studied by semiquantitative QPCR and the 2^(−∆∆Ct)^ method, using CL as reference group. A total of *n* = 3 independent experiments were performed; samples were processed in duplicate. Mean values ± SEM have been included in the graphs. Statistical analysis performed: 1- way ANOVA and Tukey’s post test (**A**–**F**,**H**) and Student’s *t*-test (**G**), results were significant when *p* < 0.05. In the figure representative images of D3 enzyme as well as TRα1- and TRβ1- receptor positive cells Hoechst 33258 stained are included (**I**–**N**). Scale bar 25 μm. Abbreviations: CL, clusters; D1, deiodinase enzyme type 1; D2, deiodinase enzyme type 2; D3, deiodinase enzyme type 3; EB, embryoid bodies; HO, Hoechst; MCT8, monocarboxylate transporter 8; MCT10, monocarboxylate transporter 10; TRα1, thyroid hormone receptor alpha 1; TRα2, thyroid hormone receptor alpha 2; TRβ, thyroid hormone receptor beta; mCL, mixed population clusters; mSC, mixed population single cells.

**Figure 5 ijms-21-08945-f005:**
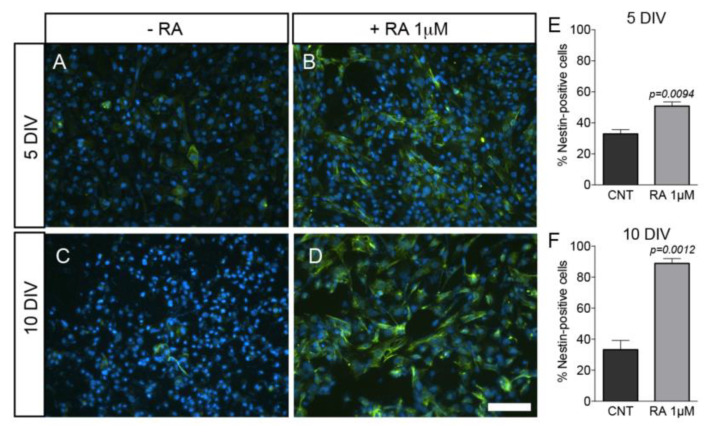
RESC and neuroectoderm lineage. Single cells obtained from the EB split were cultured onto gelatin coated coverslip in SCM medium without mitogens and allowed to differentiate. Representative images of spontaneously differentiated cells (**A**,**C**) or 1 μM RA-treated cells for 5 and 10 DIV (**B**,**D**) nestin-immunostained (green) and Hoechst 33,258 nuclear-stained cells (blue) are shown. The number of nestin-positive cells was significantly higher in the 1 μM RA-treated group of cells at both time points studied (**E**,**F**). Statistical analysis performed: Student’s *t*-test. Scale bar 100 μm. Abbreviations: DIV, days in vitro; RA, retinoic acid.

**Figure 6 ijms-21-08945-f006:**
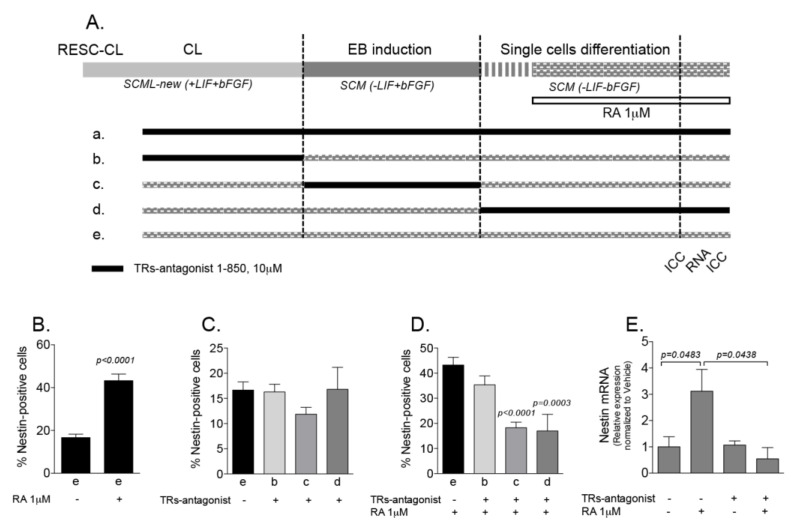
(**A**). TR antagonist 1-850 and RA treatment experimental design. The diagram shows the experimental conditions used to block T3 binding to its receptors TRα and TRβ in the different RESC culture conditions. Experimental groups: (**A**) TR antagonist 1-850 treatment for the duration of the experiment; (**B**) cells treated when growing as CL; (**C**) cells treated during the EB induction period; (**D**) cells treated during the differentiation phase as single cells obtained from EB; (**E**) control group of cells (not TR antagonist 1-850 treated). (**B**–**E**): Effect of TR antagonist 1-850 treatment in the RA-induced differentiation. (**B**–**D**): TRα and TRβ inhibition induced a reduction in the number of nestin-positive cells after 10DIV, EB and differentiating single cells obtained from EB being the RESC populations most affected. (**E**): Nestin RNA expression in single cells. RA induced an increase in the mRNA expression of nestin, which was downregulated by the TR antagonist 1-850. Graphs contain mean values ± SEM from 2–3 independent experiments performed in duplicate. Statistical analysis performed: Student’s *t*-test. Abbreviations: bFGF, basic fibroblast growth factor; CL, clusters; DIV, days in vitro; EB, embryoid bodies; LIF, leukemia inhibitory factor; RA, retinoic acid; SCML, stem cell medium with LIF; TR, thyroid hormone receptor.

**Table 1 ijms-21-08945-t001:** Sequences of primers used for real-time PCR. The name of the gene, the gene bank accession number as well as the forward and reverse primer sequence information are included in the table.

Gene	Accession Number	Primer Sequence
*Afp*	NM_012493.2	*Forward* 5′- ATGAGTAGCGATGCGTTGGC -3′ *Reverse* 5′- GTCTGGAGCGGTCTTCTTGC -3′
*D1*	NM_021653.4	*Forward* 5′- TCATTCAAGGCAGCAGACCC -3′ *Reverse* 5′- TGCCTGATGTCCACGTTGTT -3′
*D2*	NM_031720.5	*Forward* 5′- CTCCTCAGCGTAGACTTG -3′ *Reverse* 5′- GTTCCAGACACAGCGTAG -3′
*D3*	NM_017210.4	*Forward* 5′- GCATCCGCAAGCATTTCC -3′ *Reverse* 5′- GGCATCTCCTCACCTTCA -3′
*GAPDH*	NM_001113417.1	*Forward* 5′- TCATCCCTGCCTCTACTG -3′ *Reverse* 5′- TGCTTCACCACCTTCTTG -3′
*MCT8*	NM_147216.1	*Forward* 5′- CAATGGTGTGGTGTCTGC -3′ *Reverse* 5′- CGGTAGGTGAGTGAGAGC -3′
*MCT10*	XM_017601535.1	*Forward* 5′- GGTGCAGCTGTAGGATTCGT -3′ *Reverse* 5′- AGAATGACCAGTGACGGCTG -3′
*Stella*	NM_001047864	*Forward* 5′- GGTTCGTCTCCAGGTTAAG -3′ *Reverse* 5′- TCATCTCGTCTCTCATTTC -3′
*TRα* *1*	NM_001017960	*Forward* 5′- GCAAACACAACATTCCGC-3′ -3′ *Reverse* 5′- TCCTGATCCTCAAAGACCTC -3′
*TRα* *2*	NM_031134	*Forward* 5′- GCAAACACAACATTCCGC -3′ *Reverse* 5′- CACCAAACTGCTGCTCAA -3′
*TRβ*	NM_012672	*Forward* 5′- -GCAAACACAACATTCCGC -3′ *Reverse* 5′- CACCAAACTGCTGCTCAA -3′
*Nestin*	NM_001308239.1	*Forward* 5′- GGAGTGTCGCTTAGAGGTGC -3′ *Reverse* 5′- CAGCAGAGTCCTGTATGTAGCC -3′

**Table 2 ijms-21-08945-t002:** Primary and secondary antibodies. The antibodies used, the species in which they were produced, the dilutions at which they were used and the manufacturers are described in the table.

**Primary Antibodies**				
**Protein**	**Marker**	**Specie**	**Dilution**	**Company/Code**
AFP	Endoderm	*Goat*	1:50	S. Cruz Biotech./sc-130302
β-CG	Trophoblast	*Goat*	1:250	S. Cruz Biotech./sc-51605
D3	TH signalling	*Goat*	1:50	S. Cruz Biotech./sc-69388
F-Actin	Cytoskeleton	*Mouse*	1:100	S. Cruz Biotech./SC-1616
Laminin	ECM	*Rabbit*	1:80	Sigma/L9393
Nestin	Neuroectoderm	*Mouse*	1:500	BD Pharmigen/556309
OCT 4	Pluripotency	*Rabbit*	1:50	AbCam/ab19857
TR	TH signalling	*Rabbit*	1:500	Affinity Reagents/PA1-211A
TR	TH signalling	*Rabbit*	1:500	Affinity Reagents/PA1-213A
**Secondary Antibodies**				
**Target Specie**	**Host**	**Conjugate**	**Dilution**	**Company/Code**
Rabbit	*Donkey*	Rhodamine Red	1:200	Jackson Lab./711-295-152
Mouse	*Goat*	DyLight 488	1:200	Thermo Scient./35502
Goat	*Donkey*	Rhodamine Red	1:200	Jackson Lab./705-295-147

Abbreviations: AFP, Alpha fetoprotein; β-CG, Chorionic Gonadotropin beta; D3, deiodinase type 3; ECM, extracellular matrix; Jackson Lab.: Jackson Laboratories, Bar Harbor, ME, USA; OCT 4, Octamer-binding transcription factor 4; S. Cruz Biotech.: Santa Cruz Biotechnology; Thermo Scient.: Thermo Scientific; TH, thyroid hormone; TRα thyroid hormone receptor alpha-1; TRβ1 thyroid hormone receptor beta 1.

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
