# Peer review of "Thyroid Hormone Signaling in Embryonic Stem Cells: Crosstalk with the Retinoic Acid Pathway"

_ijms, 2020, doi:10.3390/ijms21238945_

Round 1

Reviewer 1 Report

In this manuscript, the authors investigated the potential role of THs and RA in early pre-implant embryonic development. To do this, they used embryonic stem cells (ESCs) derived from the inner cell mass of rat blastocyte obtained 4.5 days post coitum. Some specific culture protocols were evaluated. The analysis of nuclear receptor and co-regulator was then performed in these different conditions. From GO analysis, TH signalling emerged as one of the most represented pathways. Lastly, they demonstrated that THs played a key role in the neuroectodermal lineage induced by RA, since TR antagonists impaired RA-induced neuroectodermal differentiation. The paper is well written and data support authors' conclusions. I have only some minor comments.

Line 88. events=markers or sth else? this is unclear

Line 102. Culture medium contained 10% of serum in all aforementioned conditions (FBS-ES). Did the authors use other FBS formulations?

Line 339. 300 m ??

After TR antagonist 1-850 treatment, did the authors analyse the expression of a TR-regulated gene? 

Please provide IHC Ab codes.

Author Response

We thank the referee for valuable comments on our manuscript. 

Please, see the attached file with the reply to the Referee's point-by-point comments and questions.

Reviewer 2 Report

    Manuscript: ijms-1001422

    Suggestions/Comments:

  1. Does the non-genomic actions of TH signaling play a role in the present study? Please specify this and provide experimental evidences.

  1. Check symbols omitted.

  1. Check the grammar and phrasing of the text.

Author Response

We thank the referee for valuable comments regarding the manuscript.

The point-by-point response to the referee's comments has been included in the attached file.

Round 2

Reviewer 2 Report

I have no further comments.

Author Response

We thank the reviewer for valuable comments done in the Round 1.

The Reviewer "Comments and suggestion for authors" in this Round 2 are: "I have not further comments".